# Optimizing Precision Material Handling: Elevating Performance and Safety through Enhanced Motion Control in Industrial Forklifts

Fahim Faisal Amio [1,†], Neaz Ahmed [1,†], Soonyong Jeong [2], Insoo Jung [1,*] and Kanghyun Nam [1,*]

[1]  Department of Mechanical Engineering, Yeungnam University, Gyeongsan 38541, Republic of Korea; fahimfaisal@yu.ac.kr (F.F.A.); ahmedneaz@yu.ac.kr (N.A.)
[2]  Samdo Industry, 2557, Namhaean-daero, Goseong-eup, Goseong-gun 52936, Republic of Korea; jsyong8@naver.com
*   Correspondence: aksien@yu.ac.kr (I.J.); khnam@yu.ac.kr (K.N.); Tel.: +82-53-810-2455 (K.N.)
†   These authors contributed equally to this work.

**Abstract:** In adapting to the demands of this modernized landscape, a conventional human-operated forklift within an industrial or warehouse setting falls short. However, the adoption of autonomous forklifts remains a distant prospect for many companies, primarily due to the formidable implementation and switching costs associated with artificial intelligence and complex control mechanisms. To bridge this gap, we present the development of a teleoperated forklift utilizing mecanum wheels for enhanced maneuverability. A key contribution of this work lies in the design of a novel synchronization method for the precise position control of the pallet carriers. This method surpasses the conventional independent and master–slave approaches, demonstrably achieving superior tracking and synchronization performance. Also, a model-based velocity control algorithm was designed for the mecanum wheels to facilitate the mobility of the system. The forklift was successfully able to carry a maximum load of 300 kg. For the comparison of the tracking and synchronization performance, the independent and master–slave methods were also applied to the system. The proposed method showed better performance compared to other structures.

**Keywords:** teleoperated forklift; mecanum wheels; velocity control algorithm; alternative to autonomous forklifts; master–slave control system; tracking and synchronization performance; cost-effective teleoperated forklift; remotely operated electric forklift; mecanum wheel velocity control; industrial forklift optimization



## 1. Introduction

Forklifts have been the backbone of industry for more than 100 years—and they are being improved continuously. The loading, unloading, and transportation of materials are the chief concerns for all production facilities and largely regulate the associated costs. In a manufacturing process, the material flow, i.e., from the work site to the storehouse, is quite costly, and numerous industries aim to reduce it. The field can be categorized based on its application in various sectors, including retail and wholesale, logistics, the automotive industry, the food industry, and others. As such, it is a pressing need to establish a convenient and cost-effective system. Increasing demand for automation has resulted in significant changes in the operation of current warehouses and distribution centers. Due to the rapid advancement in control engineering, the scope of developing a remotely operated forklift is immeasurable. Consequently, numerous studies have been carried out in recent decades [1]. Initially, various research outcomes were presented regarding the development of automatic industrial forklifts [2,3]. This research explained the pallet engagement algorithm by employing a vision system. On the other hand, a different method was presented for localization and picking up the pallet by utilizing a

laser scanner [4]. However, in an autonomous system, various sensors are used to acquire such data. Thus, adding highly precise sensors with a complex control algorithm adds to the total manufacturing costs of an autonomous forklift. The objective of this initiative is to build a cost-effective mobile forklift that can be operated from a safe distance. The inclusion of higher-level automation has revolutionized the current storage and distribution centers. However, autonomous forklifts are still unattainable for many companies due to their high implementation and switching costs due to AI and the complex control mechanism. Moreover, an autonomous forklift has a high initial investment cost as well as the need for skilled operators to operate it. The recurring costs associated with maintaining and updating diverse subsystems can pose a significant financial burden for companies. This financial pressure is particularly pronounced for small- and medium-sized enterprises (SMEs), which may struggle to allocate resources for such sizable investments. However, at the same time, we cannot ignore the advantages of remote operations. To realize a user's objective or another internal state, accurate sensory information is of utmost importance. For example, vision data from the human eye have been shown to play a central role in pedestrian safety [5]. A few pros of such operations are operations in hazardous or remote environments where humans' presence is not possible and the system must therefore operate in the absence of a local human operator, such as in nuclear power stations or military environments [6]. As such, this study approaches the goal of creating a cost-effective teleoperated electric forklift capable of performing warehouse operations in an indoor environment.

In this study, the development of a remotely operated electric forklift is discussed. Firstly, a velocity control algorithm will be implemented to control the speed and direction of the mecanum wheels. Later, a focus on the use of a synchronization control strategy for the motors of the pallet carrier will be discussed. This paper emphasizes the development of a remotely operated forklift that can lift up to 300 kg and move at a speed of 2.5 km/h. The paper begins with a comprehensive description of the entire experimental system in Section 2. This is followed by two modeling sections: Section 3 details the mathematical model of the mecanum-wheeled mobile robot, and Section 4 focuses on modeling the forklift system used in the experiment. Section 5 then introduces the various position control algorithms considered for the study. The effectiveness of a chosen synchronization method is subsequently evaluated through real-time experimentation in Section 6. Finally, Section 7 presents the obtained results and their analysis, followed by the concluding remarks in Section 8.

## 2. System Description

The chassis of the forklift is a yellow-colored hollow cuboid shape with a rectangular base. The cuboid shape is preferred for storage purposes regarding the internal components. Two extended carriers are placed in front of the shell. The arms are made of cast iron, which can be used for carrying objects, up–down movements, and removing large objects or smaller packages on pallets. The teleoperated forklift possesses the capability to move promptly in all directions regardless of its current position and orientation. The wheels are affixed to the main body frame through a robust attachment method determined by damping calculations. This approach ensures that each wheel maintains surface contact with the floor, even on uneven surfaces, mitigating the impact of vibrations on the robot body. Each of the pallet carriers is mounted on a vertical robotic platform provided by Hanshin robotics. The robots are ball screw type. In addition to supporting the carriers, they ensure smooth up and down motion. The detailed technical information of the robots is provided in Table 1. Figure 1a,b show the overall structure of the forklift from the front and rear views.

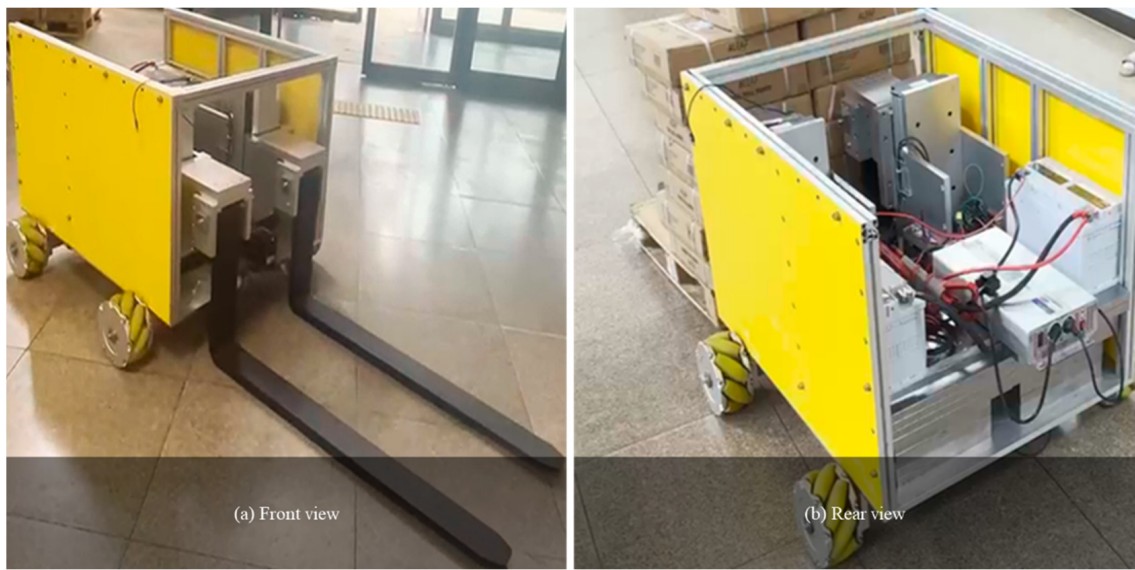

**Figure 1.** Structure of the forklift.

**Table 1.** Forklift and parts specification.

| Part | Specification | |
|---|---|---|
| Robot | Model | HR200-L70-W6B-0100R05 |
| | Type | Ball screw |
| | Length, Stroke | 200 mm, 100 mm |
| | Max load | 150 kgf (Horizontal), 50 kgf (Vertical) |
| | Max speed | 250 mm/s |
| Forklift | Capacity | 300 Kg |
| | Speed | 2.5 m/s. Max |
| | Size | 1050 × 850 × 650 (mm) |
| Battery | Nominal Voltage and capacity | 12 V, 120 Ah |
| Pallet carrier motor | Model, Type | Mitsubishi HG-KR 73(B), PMSM |
| | Rated speed | 3000 rpm |
| | Permissible load | Radial-392N, Thrust-147N |
| | Rated Torque | 2.4 Nm |
| | Rated voltage and current | 109 V, 4.8 A |
| Pallet carrier motor driver | Model | Mitsubishi MR-J4-70A |
| | Control method | Sinewave PWM control, current control |
| | Torque limit | External analog input (0 V DC to +10 V DC/maximum torque) |
| | Rated Output | Voltage 3-phase 170 V AC, current 5.8 A |
| | Communication | RS-422: 1: n communication (up to 32 axes) |
| | Encoder | Rotary incremental type, (A/B/Z-phase pulse) |

*2.1. Sytstem Setup*

2.1.1. Power Supply System

To power the system, a 12 V battery with an inverter and a power carrying capacity of 120 Ah was employed. It is noteworthy that myRIO can function efficiently with a maximum power supply of 12 V. For a comprehensive overview of all the components

comprising the forklift and their detailed specifications, refer to Table 1, which outlines each element's characteristics and specifications. This meticulous consideration ensures the harmonious integration of the components, supporting the forklift's optimal performance.

### 2.1.2. Connection Diagram

Figure 2 illustrates the experimental setup configuration of the whole system. The two pallet carriers' servo motors were controlled by the motor driver, operating in position control mode to ensure alignment with the control design framework. Meanwhile, the four mecanum wheels' BLDC motors were controlled by two O-drives using the control algorithm. The control unit for the system was the MyRIO, a microcontroller provided by National Instruments. The MyRIO functions as it receives control instructions from the upper computer base via Wi-Fi. It operates the motors through the bridge driver and transmits encoder data for feedback. This board provides an analog output of ±10 V and a precise analog input. This high precision enabled the generation of a finely tuned PWM signal for the motor input.

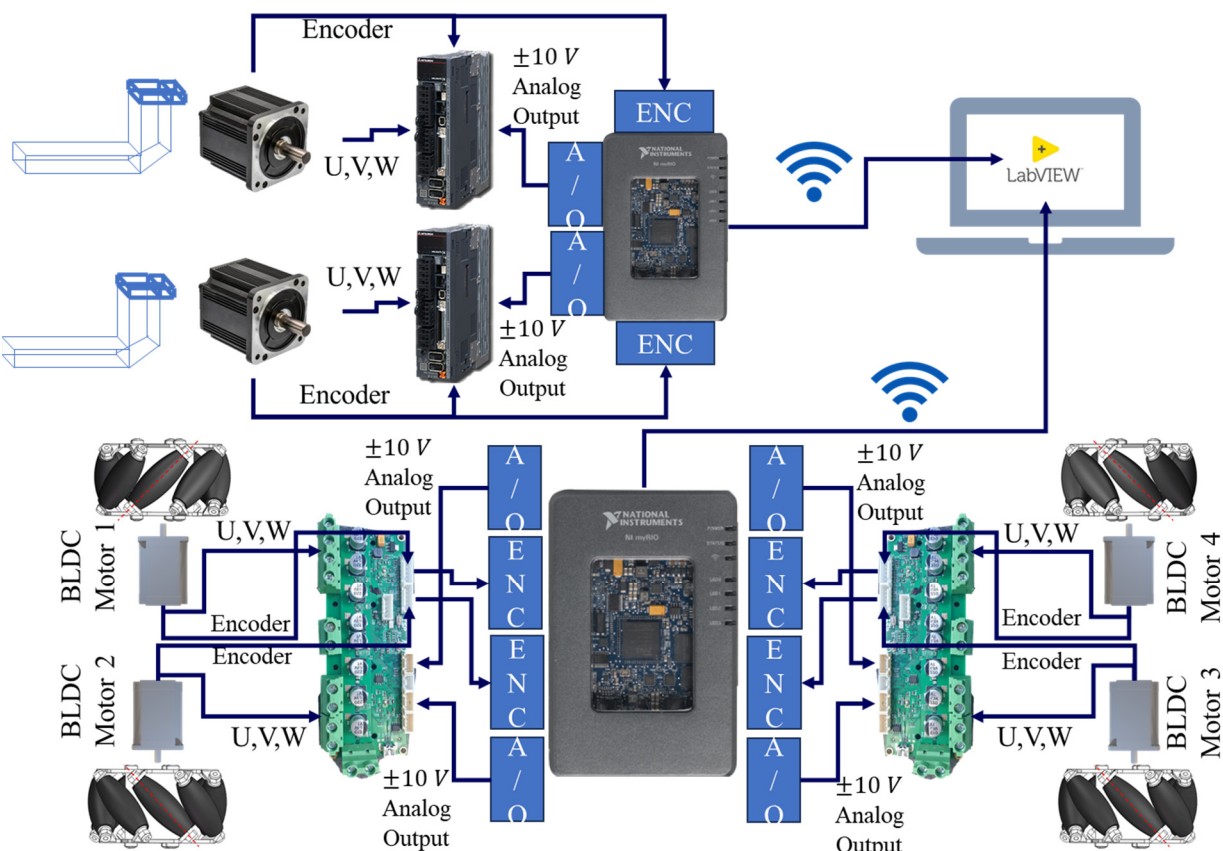

**Figure 2.** Connection diagram of the experimental system.

### 2.1.3. Control Components

MyRIO acts as a conduit for encoder data, facilitating the continuous monitoring and adjustment of the forklift's operations. This collaborative synergy between the remote PC and myRIO establishes a reliable and efficient control system.

## 3. Mecanum-Wheeled Mobile Robot System Modeling

The mobile robot is driven by four BLDC motors and attached to four mecanum wheels, which are composed of small rollers that are dispersed on the hub. These rollers possess three degrees of freedom: (a) rotation around the wheel's axis, (b) rotation around the roller's axis, and (c) rolling between the rollers and the ground. Consequently, when the wheels start rotating around the drive shaft and the rollers freely rotate around their

respective axes, the wheels are capable of actively moving in one direction while also allowing unrestricted movement in other directions. For this study, we are using a mobile platform that is equipped with four mecanum wheels. These wheels are mounted in pairs on each side of the vehicle and are positioned evenly in relation to the vehicle's center of mass (refer to Figure 3). The platform performs movement on a plane surface and encompasses three degrees of freedom, i.e., translation along the x and y axes, along with rotation about the *z* axis of the inertial frame {I}. All the definitions of the used symbol are mentioned in Table 2.

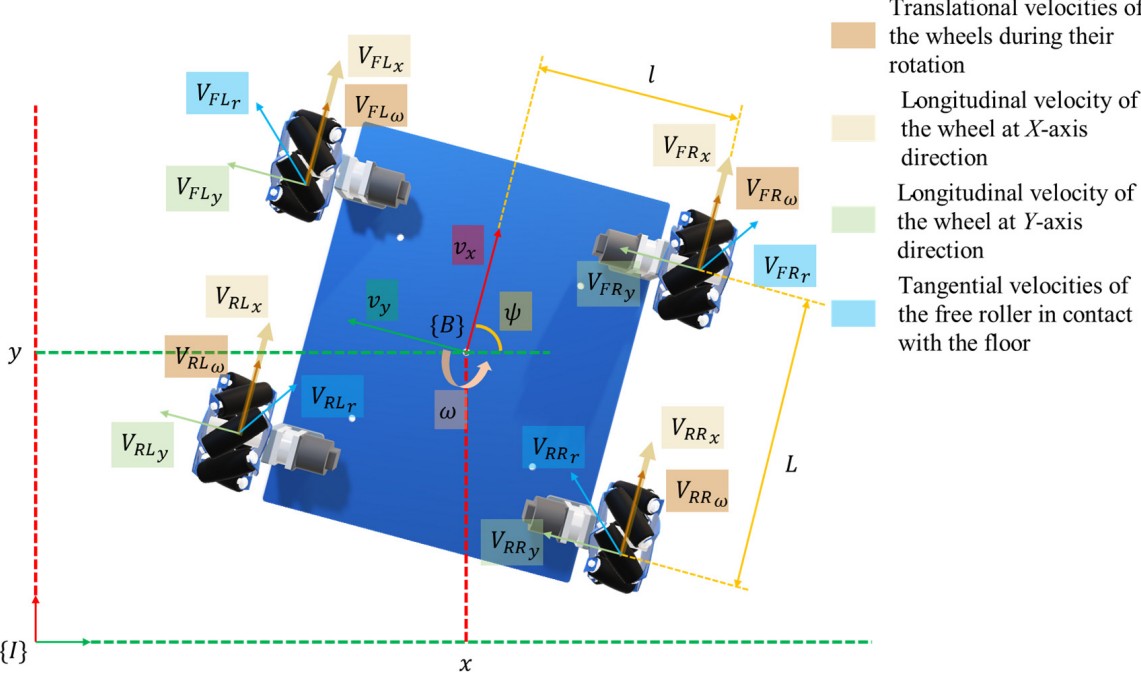

**Figure 3.** Direction of velocity vector on robot and mecanum wheels.

**Table 2.** Mecanum wheel's kinetic model variables and definitions.

| Variable | Definition |
|---|---|
| $V_{i_X}$ | Instantaneous longitudinal velocity of the wheel at *X*-axis direction [i = FL, FR, RL, RR, and these suffixes are pointed towards front left, front right, rear left, rear right wheel accordingly] |
| $V_{i_Y}$ | Instantaneous longitudinal velocity of the wheel at *Y*-axis direction [i = FL, FR, RL, RR] |
| $V_{i_\omega}$ | Translational velocities of the wheels during their rotation [i = FL, FR, RL, RR] |
| $V_{i_r}$ | Tangential velocities of the free roller in contact with the floor [i = FL, FR, RL, RR] |
| $\dot{\theta}_i$ | angular velocity of the wheels [i = FL, FR, RL, RR] |
| $v_x$ | Instantaneous velocity of the wheel along *X* axis (sway motion) |
| $v_y$ | Instantaneous velocity of the wheel along *Y* axis (surge motion) |
| $\omega$ | Instantaneous velocity of the wheel along $Z_{Wi}$ axis (yaw motion) |
| $R$ | Wheel radius |
| $\alpha$ | Angular displacement of the rollers of the mecanum wheels |
| $l$ | Lateral gap of the wheels to the center of mass |
| $L$ | Longitudinal gap of the wheels to the center of mass |
| $x$, $y$, and $\psi$ | Position and orientation of the platform in relation to the inertial frame |

### 3.1. Vehicle Kinematics

Let us assign the angular velocity of the wheels as $\dot{\theta}_i$, where i denotes FL, FR, RL, RR, and these suffixes are pointed towards front left, front right, rear left, and rear right wheel accordingly. Additionally, let us denote the velocities of the structure's center of mass in the

body frame {B} as $v_x$, $v_y$, and $\omega$. Moreover, we can define the translational velocities of the wheels during their rotation as $V_{i_\omega} = R\dot{\theta}_i$, where i denotes FL, FR, RL, RR, and $R$ represents the wheel radius. Furthermore, the tangential velocities of the free roller in contact with the floor can be expressed as $V_{i_r}$ (refer to Figure 3). Likewise, we can derive the resultant velocity of the wheels by the following:

$$
\begin{aligned}
V_{FL_X} &= V_{FL_\omega} + V_{FL_r}\sin\alpha, & V_{FL_y} &= V_{FL_r}\cos\alpha \\
V_{FR_X} &= V_{FR_\omega} + V_{FR_r}\sin\alpha, & V_{FR_y} &= V_{FR_r}\cos\alpha \\
V_{RL_X} &= V_{RL_\omega} + V_{RL_r}\sin\alpha, & V_{RL_y} &= V_{RL_r}\cos\alpha \\
V_{RR_X} &= V_{RR_\omega} + V_{RR_r}\sin\alpha, & V_{RR_y} &= V_{RR_r}\cos\alpha
\end{aligned}
\tag{1}
$$

The parameter $\alpha$ represents the angular displacement of the rollers of the mecanum wheels. In the subsequent discussion, the typical value of $\alpha$ is set to 45 degrees. Additionally, due to the fixed connection between the wheels and the platform body, the velocities of the wheels can be written with respect to the velocities of the platform body.

$$
\begin{aligned}
V_{FL_X} &= v_x - l\omega, & V_{FL_y} &= v_y + \text{L} \\
V_{FR_X} &= v_x + l\omega, & V_{FR_y} &= v_y + \text{L} \\
V_{RL_X} &= v_x - l\omega, & V_{RL_y} &= v_y - \text{L} \\
V_{RR_X} &= v_x + l\omega, & V_{RR_y} &= v_y - \text{L}
\end{aligned}
\tag{2}
$$

The symbols $L$ and $l$ represent the longitudinal and lateral interspace between the wheels and the center of mass, respectively (refer to Figure 3). By solving Equation (2) for the body velocities' X-axis direction $v_x$, Y-axis direction $v_y$, and angular velocity on Z-axis $\omega$, and substituting $V_{i_\omega} = R\dot{\theta}_i$ and $\alpha = 45°$ in Equation (1), we ultimately obtain the forward kinematics of the platform as follows:

$$
\begin{bmatrix} v_x \\ v_y \\ \omega \end{bmatrix} = \text{J}_\text{V} \begin{bmatrix} \dot{\theta}_{FL} \\ \dot{\theta}_{FR} \\ \dot{\theta}_{RL} \\ \dot{\theta}_{RR} \end{bmatrix}
\tag{3}
$$

Here,

$$
\text{J}_\text{V} = \frac{R}{4} \begin{bmatrix} 1 & 1 & 1 & 1 \\ -1 & 1 & 1 & -1 \\ -\frac{1}{L+l} & \frac{1}{L+l} & -\frac{1}{L+l} & \frac{1}{L+l} \end{bmatrix}
\tag{4}
$$

The inverse kinematics is also needed as we have to feed back the vehicle velocity for control design purposes. The inverse kinematics of Equation (3) is as follows:

$$
\begin{bmatrix} \dot{\theta}_{FL} \\ \dot{\theta}_{FR} \\ \dot{\theta}_{RL} \\ \dot{\theta}_{RR} \end{bmatrix} = J_{inv} \begin{bmatrix} v_x \\ v_y \\ \omega \end{bmatrix}
\tag{5}
$$

Here,

$$
J_{inv} = \frac{1}{R} \begin{bmatrix} -1 & 1 & (l+L) \\ 1 & 1 & -(l+L) \\ -1 & 1 & -(l+L) \\ 1 & 1 & (l+L) \end{bmatrix}
\tag{6}
$$

The non-square Jacobian matrix is represented by the term "denotes". In the end, the body velocities can be readily articulated in the inertial frame through the utilization of the aforementioned matrix.

$$\begin{bmatrix} \dot{x} \\ \dot{y} \\ \dot{\psi} \end{bmatrix} = J_I \begin{bmatrix} v_x \\ v_y \\ \omega \end{bmatrix} \tag{7}$$

The position and orientation of the platform in relation to the inertial frame can be denoted as $x$, $y$, and $\psi$.

$$J_I = \begin{bmatrix} \cos\psi & -\sin\psi & 0 \\ \sin\psi & \cos\psi & 0 \\ 0 & 0 & 1 \end{bmatrix} \tag{8}$$

### 3.2. System Identification of Mecanum-Wheeled Mobile Robot System

Within this research endeavor, a model-based control architecture will be implemented to control the system's velocity. Establishing a precise nominal model of the system constitutes the initial step. For this purpose, a chirp signal, spanning a frequency range of 0 to 30 Hz, was injected for a duration of 30 s. The ensuing output data were acquired via the motor encoder. To facilitate successive analysis, a fast Fourier transform (FFT) was utilized regarding the input–output data, yielding a Bode plot of the motor system's transfer function. Figure 4 illustrates the diagram of the system identification block [7] process, and the Bode plot in Figure 5 indicates a first-order system dynamic.

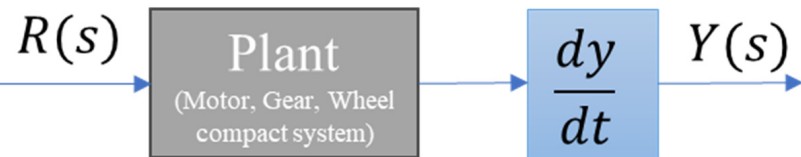

**Figure 4.** System identification process block diagram.

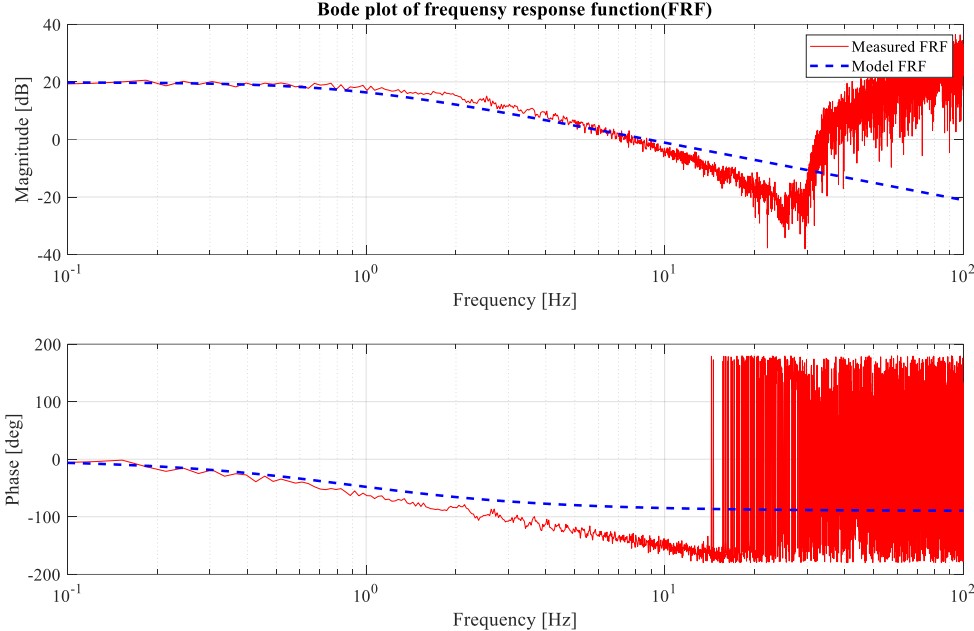

**Figure 5.** Bode plot diagram of mecanum wheeled motor system.

As we can see from the Bode plot (in Figure 5), the magnitude has dropped around 20 dB during 1 log-based frequency change, so we can be assured that the system model is 1st order [8] and should be

$$P_n = \frac{1}{J_n + B_n} \; [Here, \; J_n = 0.0180, \; B_n = 0.1019] \tag{9}$$

### 3.3. Control Design of Mecanum-Wheeled Mobile Robot System

This experiment employed a speed–voltage looped control approach for individual motor control, as described in [9]. This method posits a correspondence between applied voltage and motor speed, enabling dedicated control strategies for each actuator. The desired individual motor velocities required for achieving the overall vehicle velocity were calculated using the forward kinematics model presented in Equation (3). Subsequently, Equation (5) was used to estimate the actual velocity of each motor based on its encoder readings.

A closed-loop feedback control system was implemented for each motor, leveraging the aforementioned information. Building upon the nominal model derived in Section 3.2, which characterized the motor dynamics, a feedback controller was designed via the pole-zero cancellation method [10]. This technique aims to cancel out undesirable poles (system characteristics impacting stability and performance) by introducing corresponding zeros through controller design. A definition of all the used symbol is mentioned in Table 3.

$$C_{fb} = W_{fb} {\cdot} J_n + \frac{W_{fb} {\cdot} B_n}{s} \left( Here, W_{fb} = 2\pi \times 5 \, \text{Hz} \right) \tag{10}$$

**Table 3.** Velocity control system's related variables and definitions.

| Variable | Definition |
| :---: | :---: |
| $J_n$ | Nominal Moment of inertia (0.0180) [Related to nominal plant] |
| $B_n$ | Nominal Friction constant (0.1019) [Related to nominal plant] |
| s | Output variable for Laplace transform |
| $W_{fb}$ | Feedback band width [Related to designed control system] |
| $C_{fb}$ | Feedback control [Related to designed control system] |
| $C_{ff}$ | Feedforward control [Related to designed control system] |
| $W_{ff}$ | Feedback band width [Related to designed control system] |
| $\zeta$ | Damping ratio [Related to designed control system] |
| $W_Q$ | Q filter band width [Related to designed control system] |

This study employed a model-based feedforward controller to enhance the DC motor's speed control performance. This approach aims to proactively counteract anticipate disturbances and compensate for load variations before they impact the output (motor speed). The feedforward controller design leveraged the nominal model obtained in Section 3.2. The inverse of this model was computed to predict the control input required to achieve the desired motor speed, negating the effects of anticipated disturbances. However, to ensure system stability and prevent high-frequency control signals, a first-order low-pass filter was incorporated into the feedback loop. This filter attenuated high-frequency noise in the disturbance prediction while preserving the essential components for effective compensation. Thus, the feedforward controller's transfer function can be written as

$$C_{ff} = \frac{(J_n s + B_n) * W_{ff}}{s + W_{ff}} \left( Here, W_{ff} = 2\pi \times 40 \, \text{Hz} \right) \tag{11}$$

While feedback and feedforward control strategies yielded satisfactory performance under no load conditions, their effectiveness diminished with changing load demands. This highlights the need to account for model uncertainties inherent in the system, which can significantly impact control performance. To address this challenge, a disturbance

observer (DOB) was incorporated into the control architecture. The DOB functions as an additional feedback loop, estimating and compensating for both system disturbances and sensor noise. This enhances the robustness and adaptability of the control system, leading to improved performance across varying operating conditions.

The design of the DOB leveraged the previously obtained nominal model. The inverse of this model was utilized to predict the control input required to achieve the desired system state based on the reference signal. However, to ensure stability and robustness, a Q filter was incorporated into the feedback loop. This Q filter suppresses spurious high-frequency disturbances and sensor noise, preventing these unwanted signals from corrupting the disturbance estimate and destabilizing the control system. Also, the filter parameters dictate the dynamics of the DOB, influencing its responsiveness to different disturbance frequencies and affecting its overall impact on the control performance. The equation for the Q filter is as follows:

$$Q(s) = \frac{W_Q}{s + W_Q} \left( Here, W_Q = 2\pi \times 1 \text{ Hz} \right) \tag{12}$$

Figure 6 depicts the proposed control algorithm as a block diagram. Within this architecture, $v_x$, $v_y$, are the linear velocities in the $X$ and $Y$ axes, respectively, and the angular velocity ($\omega$) of the robot constitutes the controlled outputs.

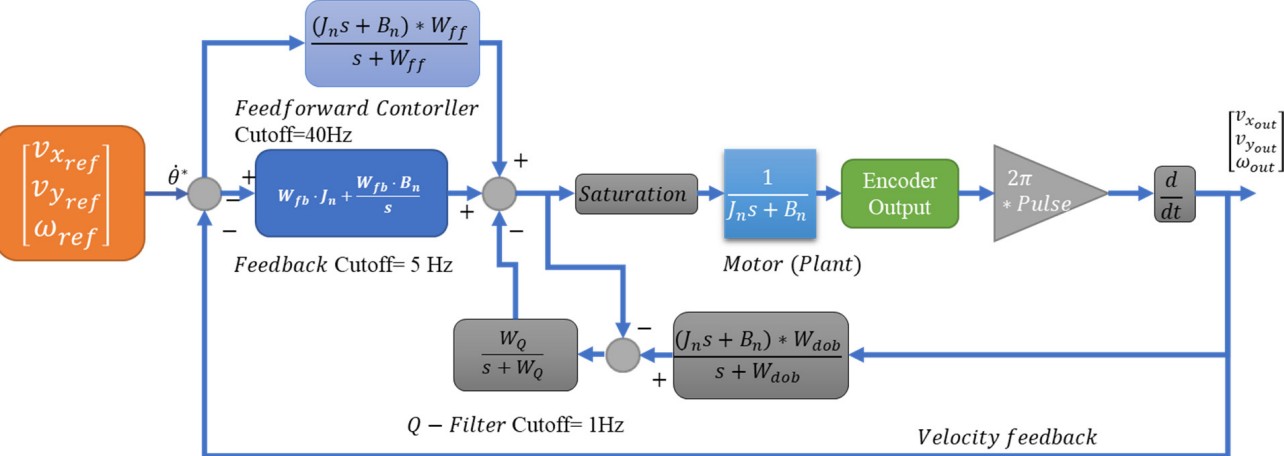

**Figure 6.** Block diagram of control algorithm.

### 3.4. Control Performance Test of Mecanum-Wheeled Mobile Robot System

The effectiveness of the proposed control system was rigorously evaluated through experimental testing. A predefined rectangular trajectory, as illustrated in Figure 7, was employed to challenge the robot's tracking capabilities under realistic conditions. To assess the system's performance, data were collected utilizing the Ultimate Stress Analysis of Tracking Robot (USATR) method [11].

Figure 8 depicts the individual wheel velocities and velocity errors during trajectory tracking. It allows for comparison of the control system's performance across each wheel, highlighting any potential discrepancies or asymmetries. Figure 9 presents the overall mobile robot's input velocity (reference trajectory), output velocity (actual tracked trajectory), and velocity error. This provides a comprehensive view of the system's tracking accuracy and the effectiveness of the implemented control algorithms. This established method facilitates comprehensive evaluation of robotic tracking accuracy and control system efficacy.

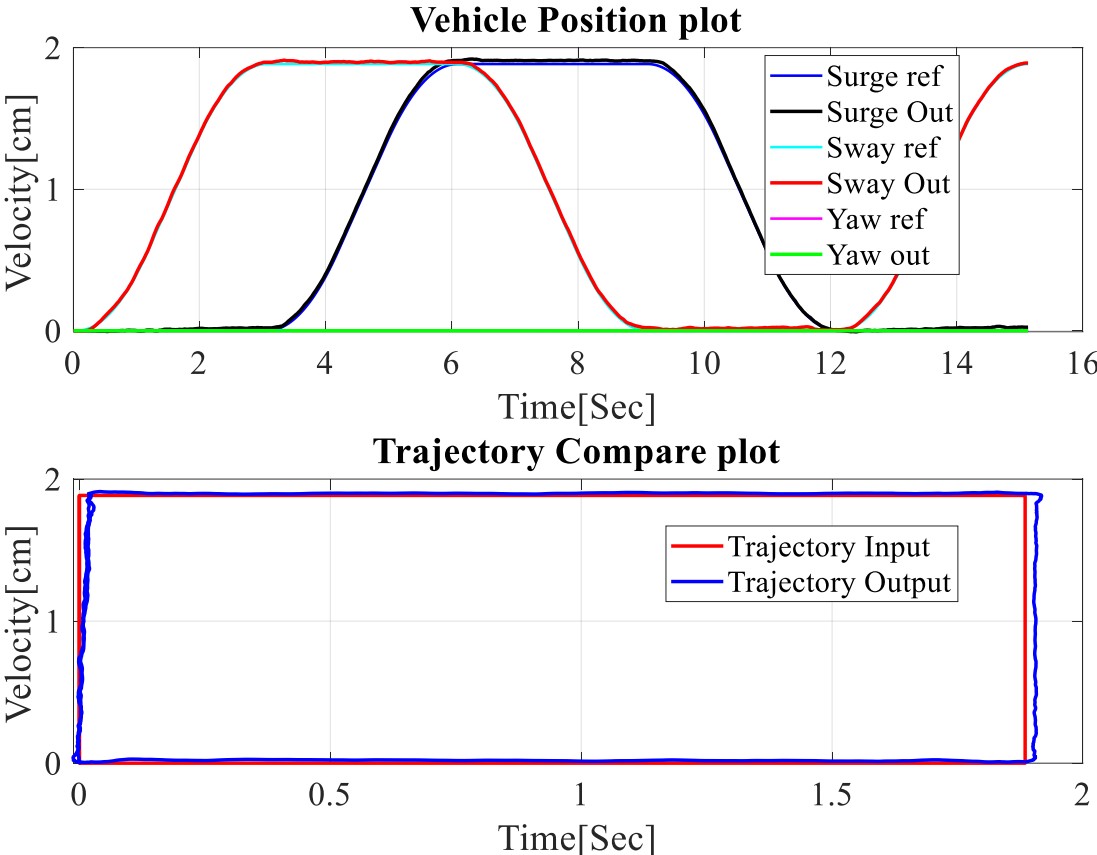

**Figure 7.** Trajectory input–output plots.

Utilizing MATLAB-2020B, the collected data were processed and visualized for detailed analysis. While the robot's instantaneous velocity could be directly determined through kinematics calculations, estimating position required further processing. To achieve this, a discrete-time integration method was applied to the velocity data. This approach allowed for accurate position estimation throughout the experiment.

Based on the error plot, we can conclude that the velocity error is below 2 cm/s on average. Although there are some overshoots at certain positions, the overall system is stable, with the absence of steady-state error.

Figure 7 depicts the performance of the control system by comparing the desired (command) position and the actual position of the robot, as measured by the motor encoder. The trajectory consists of a rectangular path, challenging the robot's tracking capabilities with direction changes and potentially demanding accelerations. The figure reveals that the robot successfully follows the command trajectory with good fidelity. The actual position closely mirrors the desired path, indicating accurate control action and effective disturbance rejection. Notably, some minor overshoot occurs at certain points, particularly during direction changes. While these deviations are negligible in magnitude, they warrant further investigation to optimize the control algorithm and minimize transient errors.

It is important to acknowledge that the current experiment did not explicitly consider wheel slip, a phenomenon where the actual wheel velocity falls short of the commanded velocity due to loss of traction. This phenomenon can significantly impact tracking performance, particularly when encountering challenging terrain or abrupt changes in direction. Therefore, investigating and incorporating mechanisms to address wheel slip remain valuable avenues for future research.

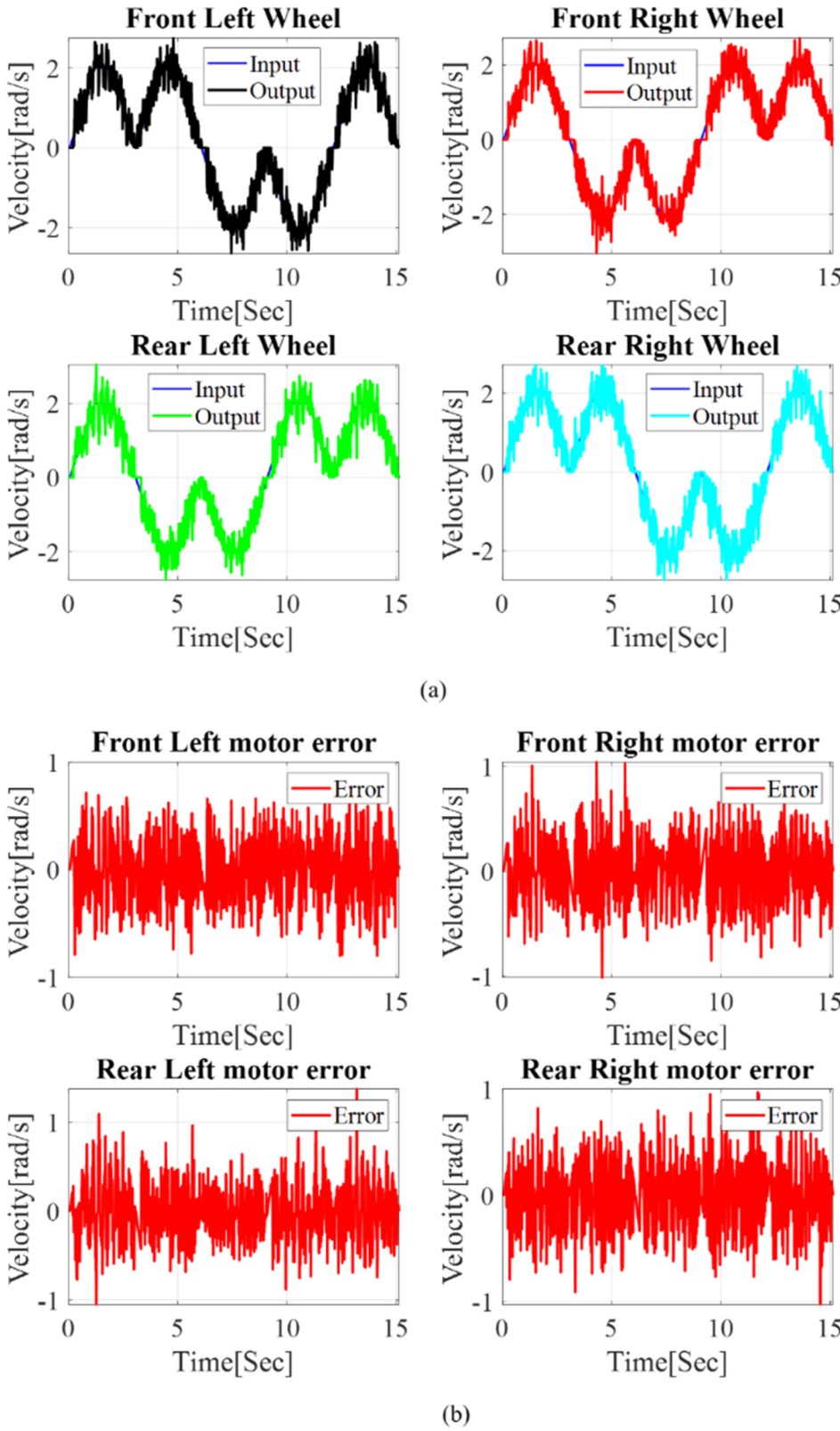

**Figure 8.** (**a**) Wheel angular velocity input–output plots. (**b**) Wheel angular velocity input–output error plots.

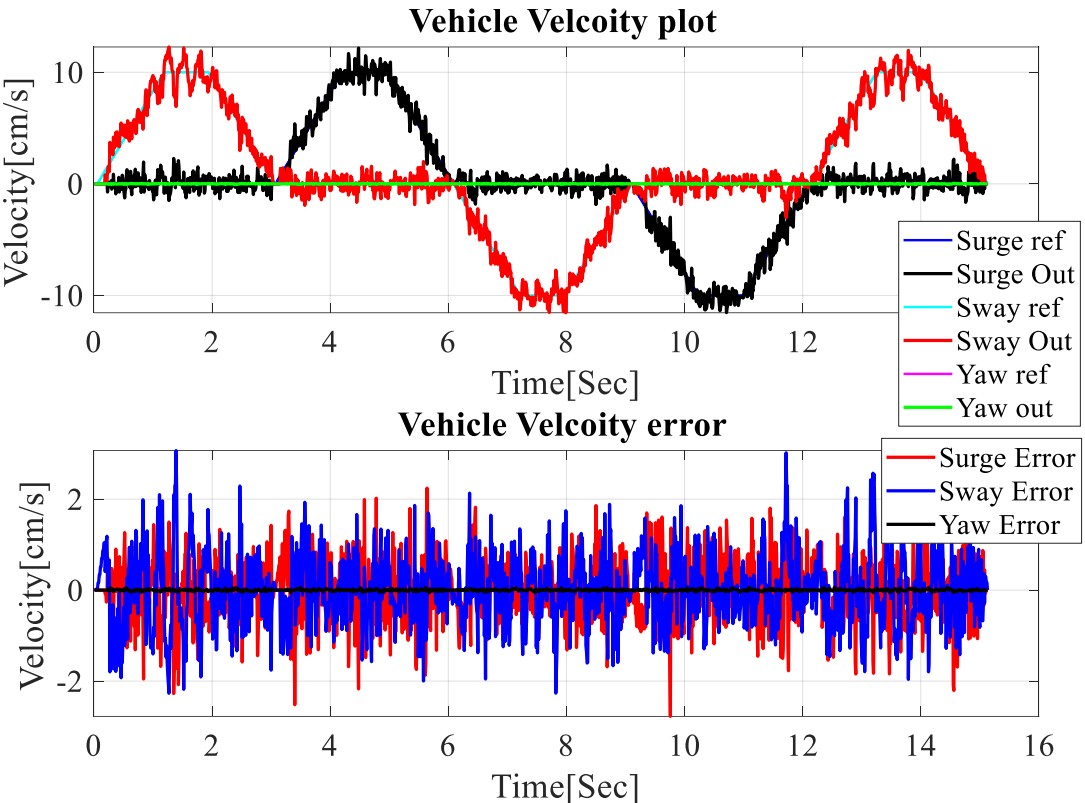

**Figure 9.** Mobile robot's input–output velocity and error plots.

## 4. Forklift System Modeling

Modeling of the forklift system is divided into two parts. Here, the position control of the pallet carrier motors will be discussed. Pallet carriers are run by servo AC motors.

### 4.1. System Dynamics of Servo AC Motor

To compute the feedback response of individual motors, it is imperative to ascertain the kinematics of both the motors. The methodology for calculating these kinematics involves a systematic process in [12] to determine the inverse kinematics first and then find the pseudoinverse [13]. Figure 10 delineates the motor parameters crucial for operation. Additionally, other variables have been taken into consideration, encompassing the following in Table 4:

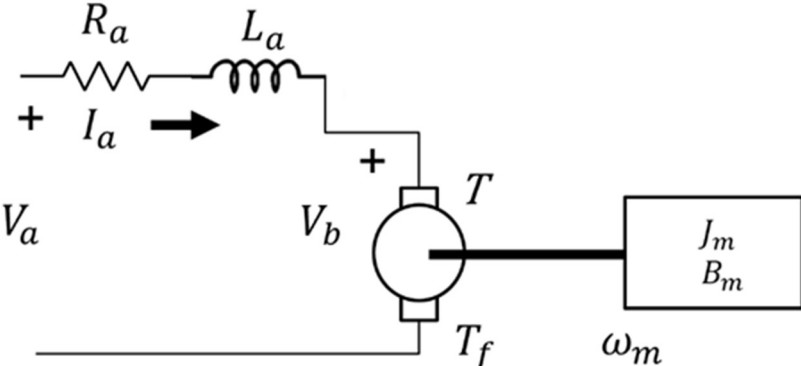

**Figure 10.** AC servo motor's parameters for operation.

**Table 4.** AC servo motor's kinetic model variables and definitions.

| Variable | Definition | Units |
|----------|------------|-------|
| $I_a$ | armature current | Amperes (A) |
| $R_a$ | armature resistance | Ohms ($\Omega$) |
| $L_a$ | armature inductance | Henrys (H) |
| $V_a$ | terminal voltage | Volts (V) |
| $V_b$ | back emf | Volts (V) |
| $\theta$ | angular position | Radians (rad) |
| $\omega_m = \dot{\theta}$ | angular velocity | (rad/s) |
| $\ddot{\theta}$ | angular acceleration | (rad/s$^2$) |
| $T$ | motor torque | Newton meters (Nm) |
| $T_L$ | load torque | Newton meters (Nm) |
| $T_f$ | frictional torque | Newton meters (Nm) |
| $J_m$ | motor inertia | Kg·m$^2$ |
| $B_m$ | motor friction co-efficient | Nm/s |

When the dynamics equation of a servo motor having one degree of motion is considered driven by an ideal motor driver, the mechanical system of a motor in time domain (t) can be defined as

$$J_m \ddot{\theta}(t) = T(t) - T_d(t) \tag{13}$$

$$\text{where } T_d(t) = T_{int}(t) + T_{ext}(t) + T_f(t) + B_m \dot{\theta}(t) \tag{14}$$

Putting the value of $T_d$ from (2) in (1) and by rearranging, we obtain

$$T(t) = J_m \ddot{\theta}(t) + B_m \dot{\theta}(t) + T_{int}(t) + T_{ext}(t) + T_f(t) \tag{15}$$

Here, $T_{int}$ is the inertial torque consisting of inertia torque and gravity effect. $T_{ext}$ denotes the external torque acting on the system. $T_f$ is the sum of Coulomb and viscosity friction.

To design a position controller, $T_{int}$, $T_{ext}$, $T_f$ will be considered as system uncertainties, and nominal values of $J$ and $B$ of the motor can be written as

$$Jn = J \cdot m J_n = J_m - \Delta J \tag{16}$$

$$B_n = B_m - \Delta B \tag{17}$$

where $J_n$, $B_n$ $\Delta J$, $\Delta B$ represent the nominal and uncertain values of $J$ and $B$. Equation (15) can be rewritten in terms of nominal model of the plant as

$$T(t) = J_n \ddot{\theta}(t) + B_n \dot{\theta}(t) \tag{18}$$

The uncertainties of the system will be dealt with by designing a proper model-based feedback controller and disturbance observer. Additionally, a feedforward controller will be added to compensate for the time delay of the system. By taking transfer function of Equation (18) and rearranging, the relationship between output angle ($\theta$) and motor torque ($T$) can be written as

$$\frac{\theta(s)}{T(s)} = \frac{1}{J_n s^2 + B_n s} \tag{19}$$

*4.2. System Identification of the Servo AC Motor*

Given that electrical components like motor resistance and inductance are controlled by the motor driver, our attention will be directed towards the mechanical parts for system identification. For system identification [7], the pallet carriers were considered as parts of the plant. Model matching method was implemented using System Identification Toolbox (SIT) in MATLAB. The nominal plant matched 95.32%. A step signal of 0–10 Hz was applied for 10 s.

## 5. Position Controller Design

The method for motor control [14] used in this experiment was position–voltage looped control. Voltage was considered input and position as output and controllers were designed for each individual motor. Then, from Equation (19), we can calculate the actual position provided by each motor encoder. Then, using the provided position and actual position, we can design feedback control loop. As we have a nominal model from system identification, we have designed feedback control loop PID controller through pole placement method. The feedback control design is as follows:

$$C_{fb} = \frac{k_p s + k_i + k_d s^2}{s} \tag{20}$$

To alleviate the loading torque on the AC motor position, we implemented a feedforward compensation strategy. The control strategy employed the design of a feedforward control law. This law was derived by inverting the nominal system model and subsequently applying a low-pass filter. The feedforward control equation for this purpose is as follows:

$$C_{ff} = \left( J_n s^2 + B_n s \right) \times \frac{w_c^2}{s^2 + 2\zeta w_c s + w_c^2} \tag{21}$$

While feedback and feedforward control proved sufficient for a no load condition, we observed a drop in the control system's performance with changes in the load. Model uncertainty became a significant factor in these scenarios. To address this issue, we introduced a disturbance observer [15]. This component handles system disturbances, sensor noise, and contributes to an overall improved control system performance. To design the disturbance observer, we utilized the inverse of our nominal model with a *Q* filter. The equation for the *Q* filter is as follows:

$$Q(s) = \frac{\omega_q^2}{s^2 + 2\zeta \omega_q s + \omega_q^2} \tag{22}$$

*5.1. Synchronization Methods for Pallet Carrier Motors*

Various types of synchronization control strategies for dual/multi-motor systems have been thoroughly investigated over the past few decades [16]. For this study, a new synchronization method has been proposed and a comparison of performance was performed with respect to the independent and master–slave control methods.

### 5.1.1. Independent Control Method

In the independent mode, all motors in the control system take a common input signal so that each motor has a unified output [17]. This control strategy provides satisfactory stability and dynamic performance. The diagram of this structure is demonstrated in Figure 11. Here, u is the desired signal, and $y_1$, $y_2$ are outputs of motor 1 and motor 2, respectively. Additionally, $d_1$, $d_2$ are disturbances. All the definitions of used symbols are mentioned in Table 5.

**Table 5.** Motor's kinetic model variables for position control.

| Variable | Definition | Unit |
|---|---|---|
| $J_n$ | Nominal MOI $(0.0073969)$ (from system identification) | Kg·m$^2$ |
| $B_n$ | Nominal Friction constant $(0.43571)$ (from system identification) | Nm/s |
| $W_{fb}$ | Feedback band width | Hz |
| $C_{fb}$ | Feedback control | - |
| $C_{ff}$ | Feedforward control | - |
| $W_{ff}$ | Feedback band width | Hz |
| $\zeta$ | Damping ratio | - |
| $W_Q$ | Q filter band width | Hz |
| $x_d$ | Target position input | rad |
| $x_2$ | forklift motor-1 (master) position output | rad |
| $x_3$ | forklift motor-2 (slave) position output | rad |
| $x_f$ | Position feedback for contact force | Nm |
| $k_1$ | Sync error gain | rad |
| $k_2$ | Tracking error gain | rad |
| $x_d - x_2$ | Tracking error of master motor | rad |
| $x_2 - x_3$ | Sync error between master and slave | rad |
| $x_d - x_3$ | Tracking error of slave motor | rad |

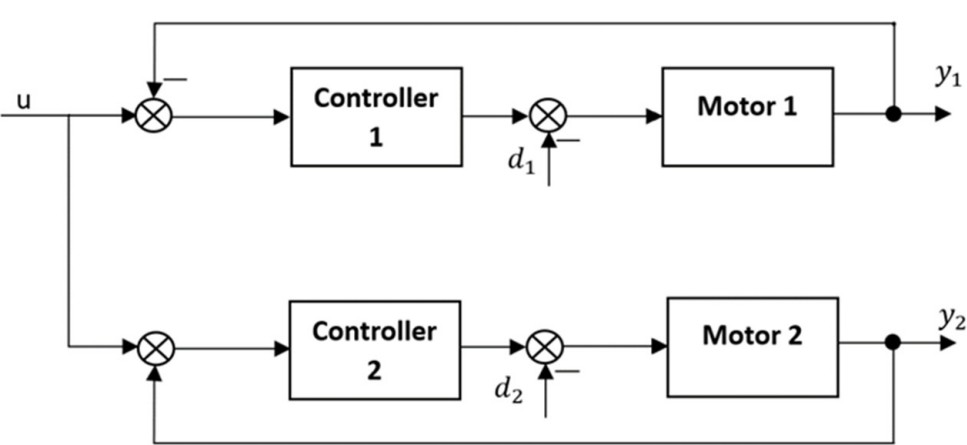

**Figure 11.** Block diagram of independent control method.

5.1.2. Master–Slave Control Method

In this configuration, the output of the master is provided as the input reference to the slaves. Consequently, the slaves will follow any velocity perturbations or load disturbances experienced by the master, thus achieving synchronous and coordinated control. However, disturbances in a slave will neither be sent back to the master nor to any other slave [18]. Here, the response of the master motor control loop is slower than the slave. The architecture of this system is illustrated in Figure 12.

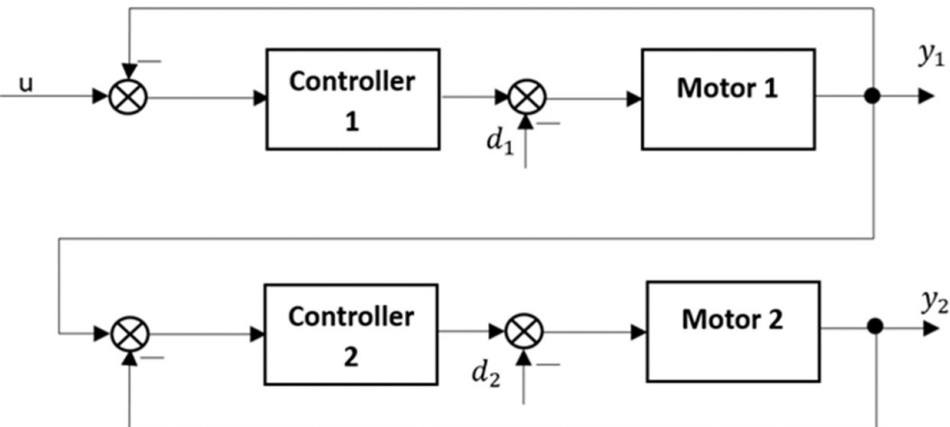

**Figure 12.** Block diagram of master–slave control method.

### 5.1.3. Proposed Control Method

This paper introduces a new synchronization control method to improve the master–slave synchronization control. The improved method to design the dual-motor synchronization control system and a proportional regulator with feedforward compensation were used. The synchronization control system designed is shown in Figure 13 as a block diagram. Here, the master follows the desired signal, whereas the slave follows the master's output. Furthermore, synchronization error is fed back to master and tracking error is fed back to slave.

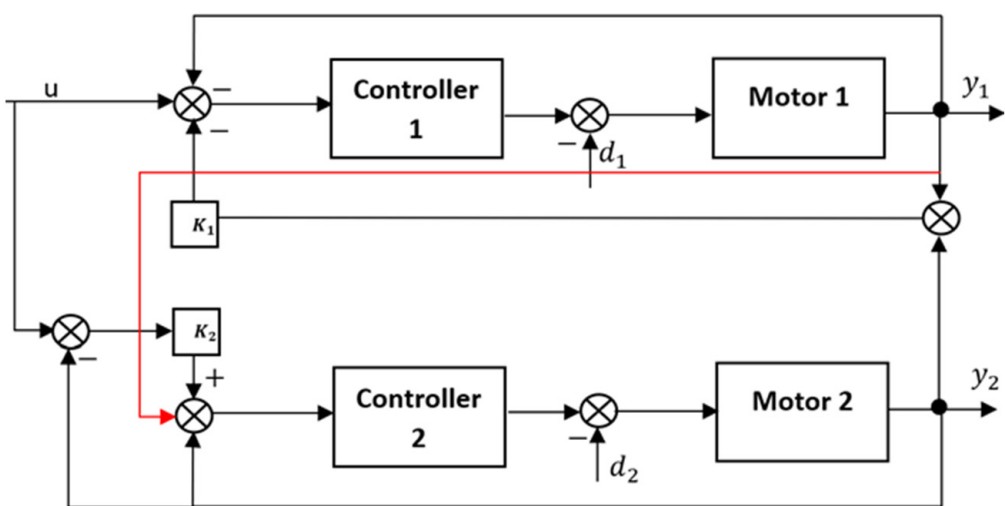

**Figure 13.** Block diagram of proposed control method.

As a result, both tracking and synchronization are compensated for in the overall system regarding both master and slave sides. In the diagram, $k_1, k_2$ are the synchronization and tracking error gain, respectively.

Figure 14 shows the design of the control algorithm where positions of the forklift motors are controlled with feedforward and feedback loop along with Disturbance Observer while being run in proposed synchronized mode. We can write controller-1 and plant of motor-1 as $G_m(s)$, $P_m(s)$, respectively, for the master side, and controller-2 and plant of motor-2 as $G_s(s)$, $P_s(s)$, respectively, for the slave side. Output of master and slave $y_1(s)$, $y_2(s)$ can be written as Equations (23) and (24)

$$y_1(s) = (u(s) - k_1 e_s(s)) * G_m(s) \qquad (23)$$

$$y_2(s) = (y_1(s) - k_2 e_t(s)) * G_s(s) \qquad (24)$$

$$\text{where } e_s(s) = y_1(s) - y_2(s)$$

$$e_t(s) = \mathrm{u}(s) - y_2(s)$$

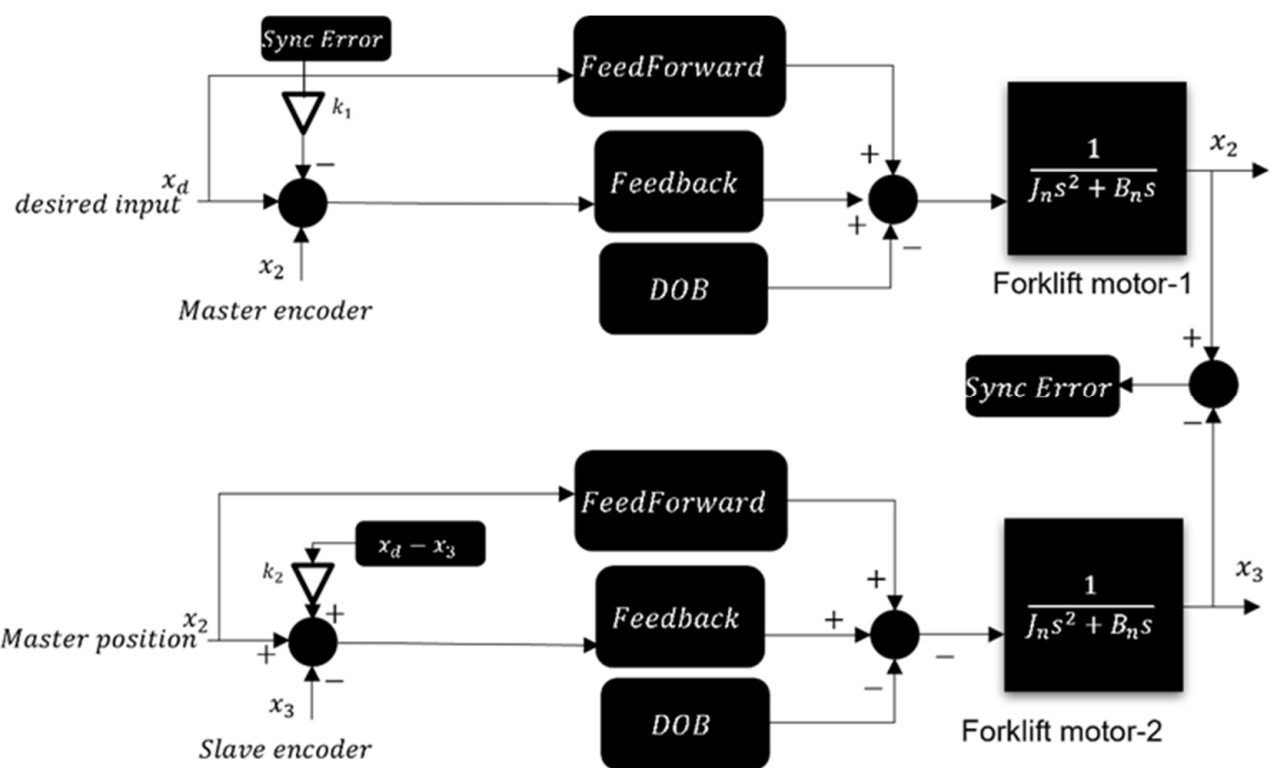

**Figure 14.** Design of proposed control method.

$k_1, k_2$ are gains that were selected by trial-and-error method during experiment.

## 6. Experiments of Synchronization Performance

### 6.1. Experimental Setup

As shown in Figure 2, at the heart of the control system lies the myRIO microprocessor, renowned for its high-resolution analog input–output capabilities. Within this framework, myRIO takes on the role of the main controller, receiving control instructions from the computer base via Wi-Fi. It regulates the AC motors through the motor driver, ensuring precise and responsive control over the forklift's movements.

### 6.2. Experiments

The performance of each synchronization method [19] was evaluated in terms of no load, 200 kg equally distributed load, and an uneven load condition where motor 1 was placed with 60 kg and motor 2 was placed with 140 kg of load. Figures 15–17 show the performance [20] of the synchronization methods for the no load, even, and uneven load conditions, respectively.

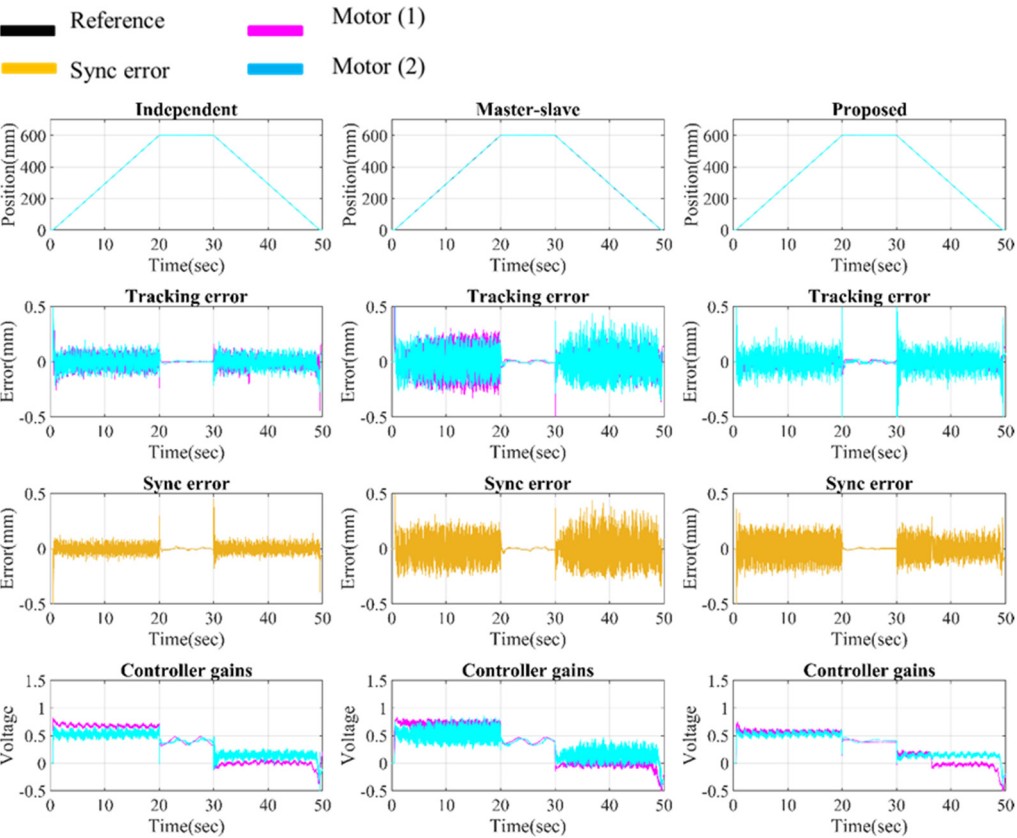

**Figure 15.** No load condition results.

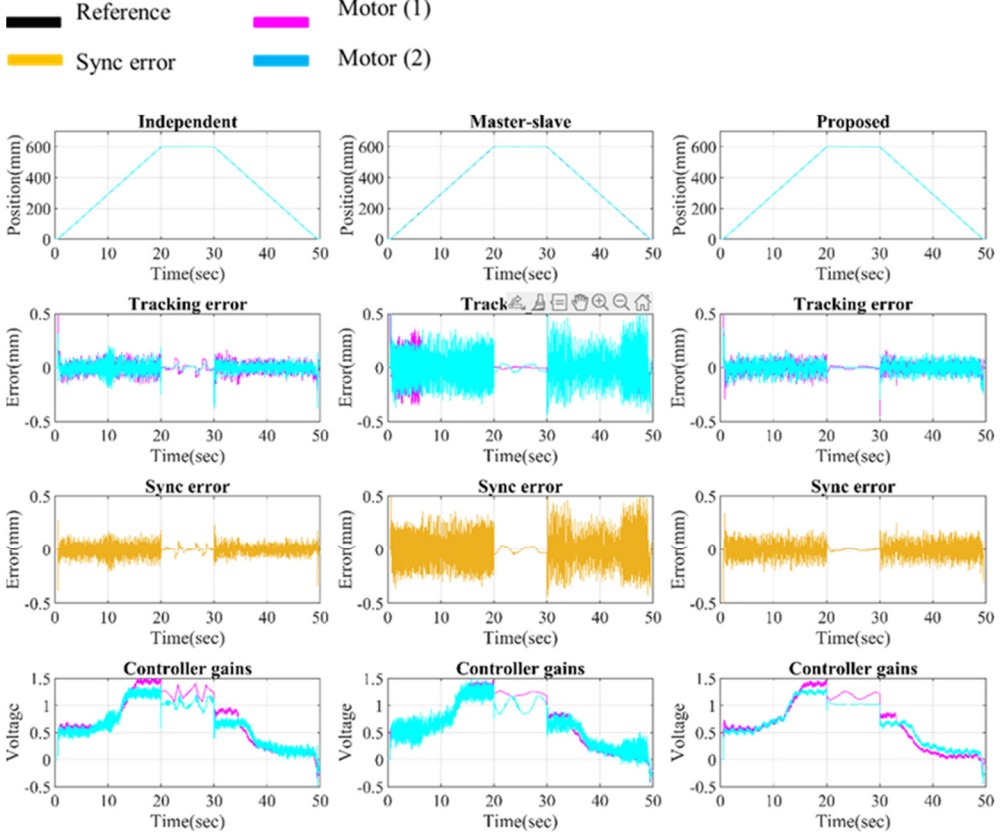

**Figure 16.** Equal load condition results.

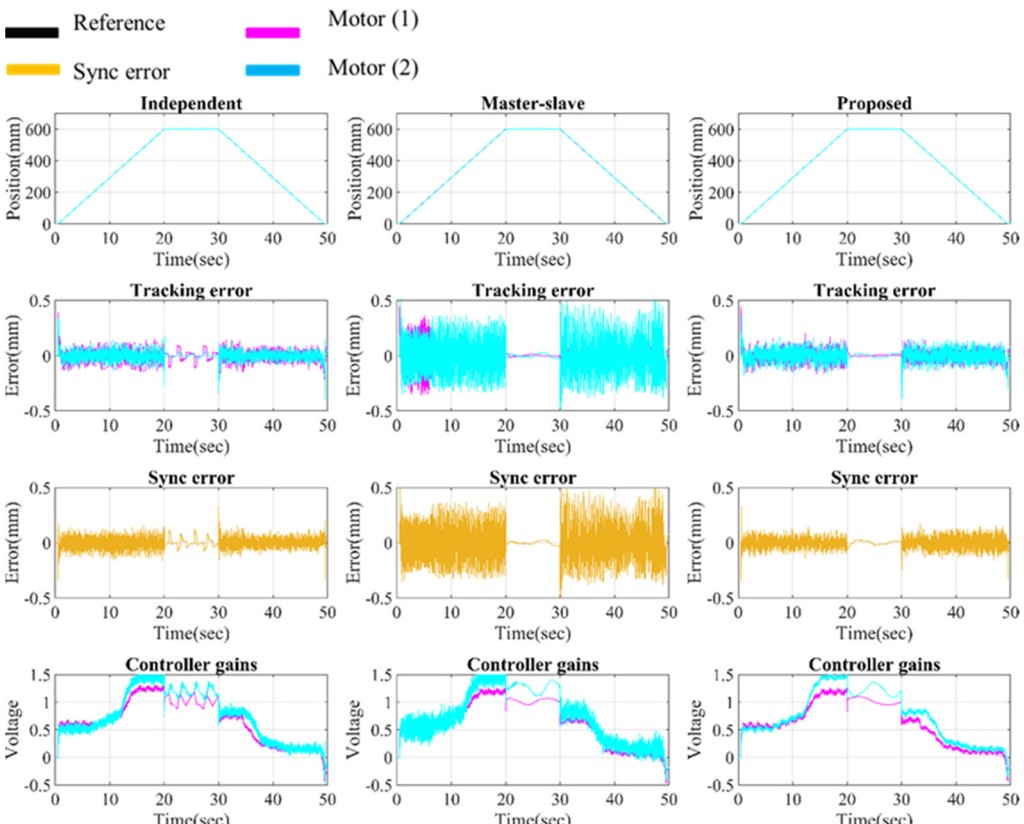

**Figure 17.** Unequal load condition results.

### 6.2.1. No Load Condition

From the experimental result demonstrated in Figure 15, it is evident that, when the system was operating without any load all the synchronization methods provided acceptable performance. However, it is evident that the independent method's performance and the proposed method's performance were superior to master–slave mode in terms of error. From the figure, it is evident that the proposed method has fewer voltage fluctuations compared to the other methods throughout the operation.

### 6.2.2. Equal Load Condition

Figure 16 illustrates the experimental condition when 100 kg of load was placed on each of the pallet carriers of the system and 200 kg in total. The independent method shows significant fluctuations in tracking and synchronization error when the load was held on the top position of the system. Consequently, control voltage also fluctuated to provide the required torque on those points. For the master–slave method, fluctuations can be observed at the top as well, but the occurrence rate is lower than in independent mode.

In evenly distributed load condition, for master–slave method, fluctuations can be observed at the top as well, but the occurrence rate is lower than in independent mode. However, the overall tracking and synchronization error is quite high in the overall operation. On the other hand, the proposed method shows a smooth curve throughout the operation in terms of both tracking and synchronization performance. The load was stable when held at the top.

### 6.2.3. Unequal Load Condition

For this experiment, a total of 200 kg load was placed on the system, same as the previous condition. However, it differs from the previous condition such that 60 kg load was placed on one side and 140 kg load was placed on the other pallet carrier. Thus, each motor was subjected to a different amount of load. Maintaining an optimal synchronization

between the position of the motors in this condition is the true testament to depicting how well-suited a system is for ensuring a desired synchronization performance. Figure 17 indicates each synchronization method's performance. For the independent method, frequently occurring conspicuous fluctuations for both synchronization and tracking error can be observed. Furthermore, master–slave shows the lowest performance when handling unequal amounts of load. Large synchronization and tracking error are evident with large number of voltage fluctuations as a consequence. Meanwhile, the proposed system handled unequal load like a champ. As mentioned above, unequal load situation is the true test of synchronization performance of a system.

## 7. Results and Analysis

Firstly, to obtain a clear understanding of the performance, the errors will be presented as RMS values in Figure 18. Secondly, how much less synchronization error occurred in the proposed method compared to the other two methods will be represented in terms of percentage in Figure 19.

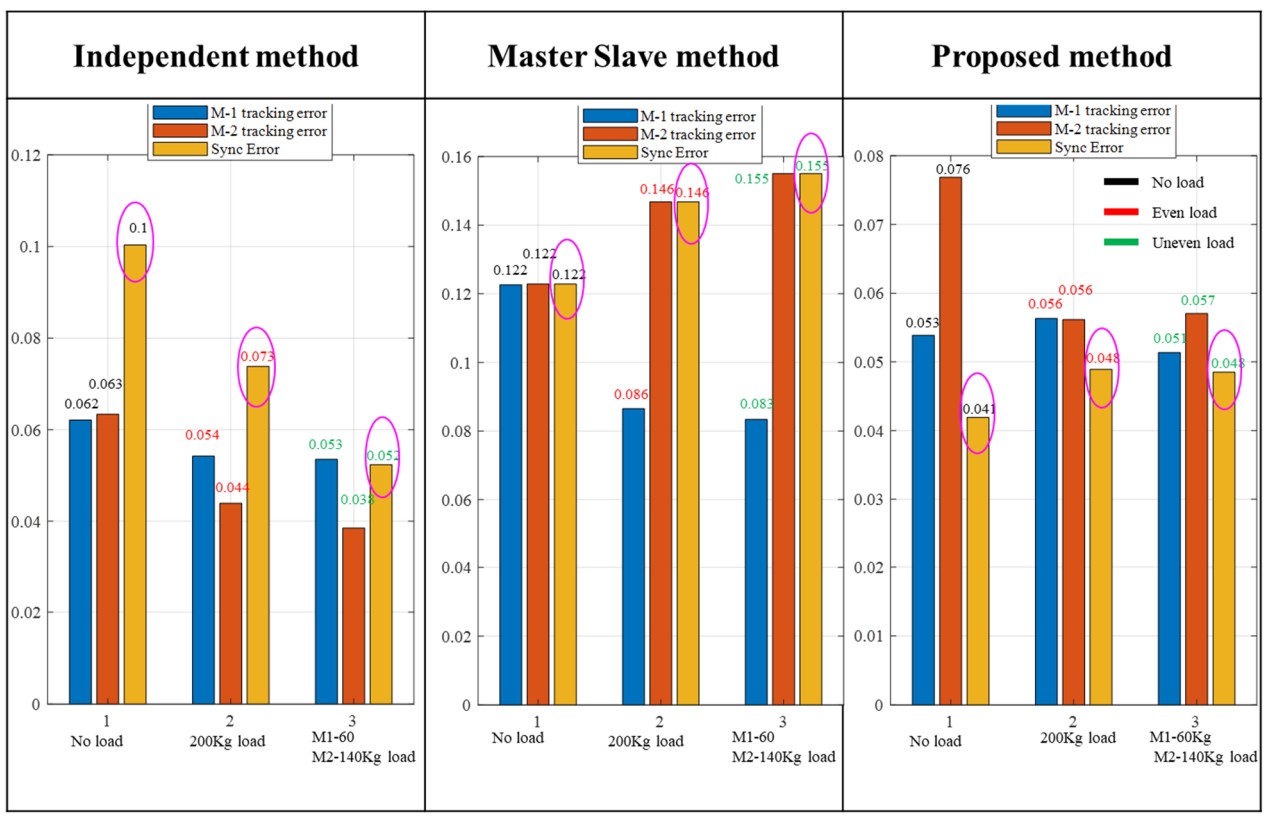

**Figure 18.** RMS errors analysis of the experiments.

### 7.1. RMS Analysis of Errors

7.1.1. RMS Error during No Load

When the system was operating without any load, the independent and master–slave methods faced rather similar amounts of sync error, i.e., 0.1 and 0.122 mm, respectively. However, the independent mode showed less tracking error compared to master–slave. Meanwhile, the proposed method was subjected to only a 0.41 mm sync error.

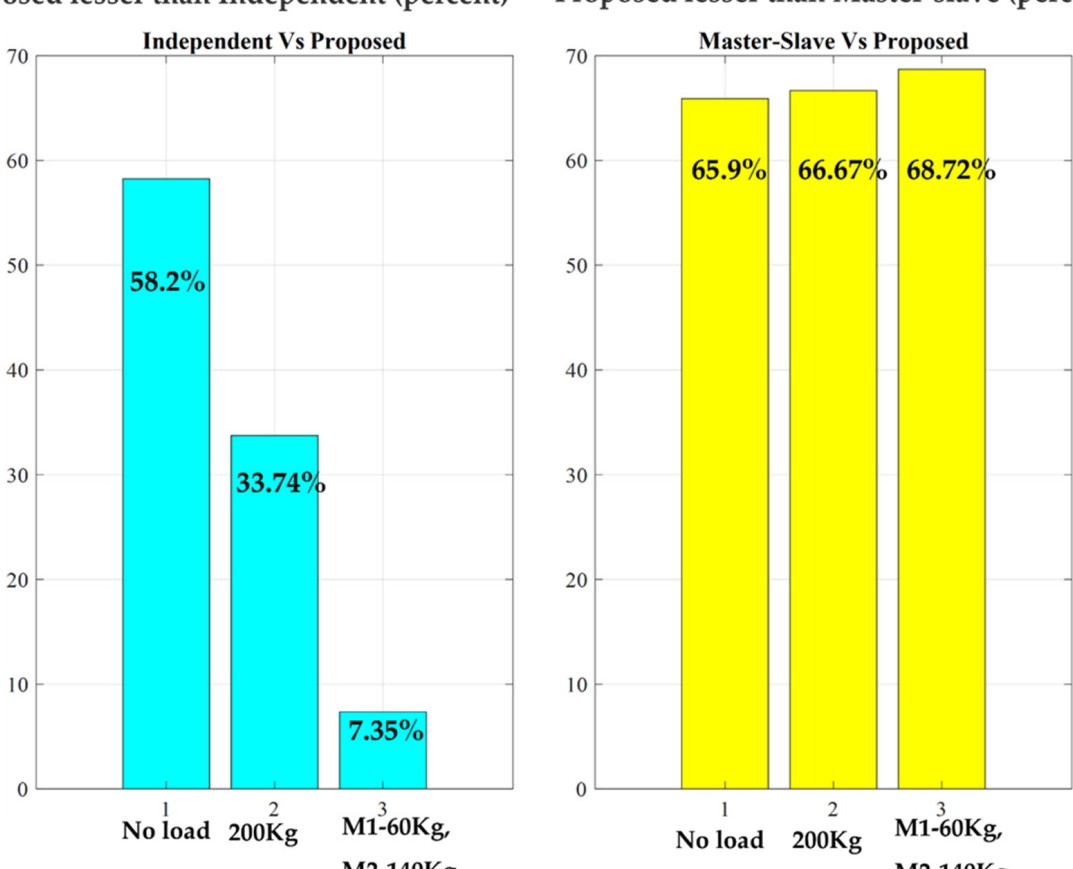

**Figure 19.** Percentage analysis of the experiments.

7.1.2. RMS Error during Equal Load

In this experiment, the highest value of rms sync error can be observed in the master–slave mode. The independent mode showed more promise, with a 0.073 mm sync error. However, the proposed method depicted the best performance with the lowest RMS sync error of 0.048 mm. Similar to the sync error, master–slave also encountered the highest amount of tracking error. Although the independent method showed less tracking error, the proposed method is not too far off, with the error being in the acceptable range.

7.1.3. RMS Error during Unequal Load

As mentioned above, an unequal load situation is the true test of the synchronization performance of a system. So, following the previous trend of showing the most promising synchronization performance, the proposed method showed a 0.048 mm error, which is the same as the equal load result. The independent mode was the second-best option with a 0.052 mm error. The master–slave mode indicated the highest amount of error of 0.155 mm for this condition. In terms of the tracking performance, the independent mode was only able to achieve a better performance regarding one motor. The difference in the tracking performance between the motors is the main reason for the poorer synchronization of the independent mode compared to the proposed method.

*7.2. Percentage Analysis of Errors*

A percentage analysis is presented in Figure 19 to obtain a broader picture of the improvement impact. For the no load situation, the proposed method's sync error is 58.2 and 65.9% less than the independent and master–slave modes, respectively. While handling a 200 kg equally distributed load, the proposed structure maintained its dominance over the

other two with 33.74 and 66.67% less error. Finally, during the unequal load experiment, 7.35 and 68.72% less sync error was evident. So, from the analysis of the results, it is evident that the proposed method retained optimal performance for all the types of load conditions that may occur during actual load conditions in warehouses or material handling operations.

## 8. Conclusions

In this research, we have designed and applied control algorithms for a cost-effective remotely operated small-scale industrial forklift. Herein, the velocity control algorithm was used for the mecanum wheels, and a synchronized position controller was employed for the precise lifting of loads by the pallet carriers. The system was capable of efficiently carrying up to a 300 kg load. To carry out this research, an extensive literature review was conducted to find an optimal synchronization method because the system used two separate motors to operate the forklift. Indeed, this is the core difference and most significant feature of this forklift compared to its peers. Controlling two separate motors provided more flexibility and reliability in handling uneven loads. Mechanically coupled pallet carriers moved by a single motor are subject to movement restriction. So, if one carrier is experiencing a greater load than the other, there is no means available to compensate for the extra load; as such, it could lead to failure of operation due to falling objects. That is why the core focus of this research was on refining the synchronization control for forklift operations, a critical factor in ensuring optimal functionality. The proposed synchronization method successfully retained the synchronization and tracking performance during all the load conditions. As a future initiative to continually improve the remotely operated forklift, a haptic feedback system will be implemented. This addition aims to further enhance safety during operation, offering operators haptic sensations of the forklift's interactions with its environment. Based on this, the operator will receive real-time information regarding the operation and will be able to take corrective action. This research holds crucial significance for small-scale material handling operations with respect to the cost-to-performance ratio.

**Author Contributions:** F.F.A., N.A. and K.N. took the lead in writing the paper, developing control algorithms, and conducting experiments. S.J. and I.J. reviewed the overall content and supervised the control development. All authors have read and agreed to the published version of the manuscript.

**Funding:** This work was supported by the 2021 Yeungnam University Research Grant (No. 221A380006).

**Institutional Review Board Statement:** Not applicable.

**Informed Consent Statement:** Not applicable.

**Data Availability Statement:** This study did not report any data.

**Conflicts of Interest:** Author Soonyong Jeong was employed by the company Samdo Industry. The remaining authors declare that the research was conducted in the absence of any commercial or financial relationships that could be construed as a potential conflict of interest.

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
