# Peer review of "Optimizing Precision Material Handling: Elevating Performance and Safety through Enhanced Motion Control in Industrial Forklifts"

_electronics, doi:10.3390/electronics13091732_

Round 1
Reviewer 1 Report
Comments and Suggestions for Authors
This paper demonstrate the design of a tele-operated forklift and its control methods. The effectiveness of the system has been showed through various experiments. There are some concerns need to be addressed before publish.
- The system contains several different components and control methods. It will be helpful to provide a description of the whole paper arrangement at the end of Introduction.
- Structure of this paper need to be improved. For example, some repeated contents appears in different sections, please reorganise these.
- Mathematical presentation need to be improved. For example, line 144, I believe it should be \dot{\theta}_{i}, instead of \dot{\theta}_{\iota}, also for the following sections where \theat appears; for equation (3) and (5), the derivative dot should be at the top of \theta, the L and R need to be at the subscript.
- Although a table of math notations has been provided, please also provide more details at the first appearance of these math representations. For example, line 146, instead of claiming all v_x, v_y, and \omega as velocities, providing definition of each velocity.
- Reference citations need to be fixed for Section 6.
- Please use the same scale for the subfigures of Figure 18. to improve the comparison view. Same comment for Figure 19.
Comments on the Quality of English Language- Abstract can be improved by summarising the experiment results.
- Many repeated contents appeared in line 81, line 96, and line 121.
Author Response
Dear Editor,
We appreciate the opportunity to revise and resubmit our manuscript, allowing us to address the insightful comments provided by the reviewers.
In this resubmission, we have included the following supplementary materials:
(a) A point-by-point response to the reviewers' comments (provided below), hereafter referred to as the "response to reviewers."
(b) An updated version of the manuscript with revisions highlighted in yellow.
Best regards,
Kanghyun Nam

Reviewer 2 Report
Comments and Suggestions for Authors
1. This paper designs and applies a control algorithm for an efficient remote-operated small-scale industrial forklift.
2. To accurately lift the load of the pallet carrier, lifting was performed using a speed control algorithm on a position controller synchronized with the Mecanum wheel.
3. Controlling two separate motors also provided more flexibility and reliability when handling uneven loads.
4. This study can be considered an excellent paper as it has important implications for small-scale material handling operations with regard to cost-performance ratio.
However, please correct the following.
1. Enlarge Figure 2, 15, 16, 17, 18
2. Add content to Abstrat to highlight the excellence of this study
3. Add explanation for definitions in Tables 3, 4, and 5
This paper complies with the format and procedures of a thesis and has various differences, so a minor revision is granted.
Author Response

(The authors gave the same response as above.)
